# Evaluation of a Probability-Based Predictive Tool on Pathologist Agreement Using Urinary Bladder as a Pilot Tissue

**DOI:** 10.3390/vetsci9070367

**Published:** 2022-07-18

**Authors:** Emily Jones, Solomon Woldeyohannes, Fernanda Castillo-Alcala, Brandon N. Lillie, Mee-Ja M. Sula, Helen Owen, John Alawneh, Rachel Allavena

**Affiliations:** 1School of Veterinary Science, The University of Queensland, Gatton, QLD 4343, Australia; s.woldeyohannes@uq.edu.au (S.W.); r.allavena@uq.edu.au (R.A.); 2School of Veterinary Science, Massey University, Palmerston North 4100, New Zealand; f.castillo-alcala@massey.ac.nz; 3Ontario Veterinary College, University of Guelph, Guelph, ON N1G 2W1, Canada; blillie@uoguelph.ca; 4Charles River Laboratories, 54943 N. Main Street, Mattawan, MI 49071, USA; mee-ja.sula@crl.com; 5QML Vetnostics, 11 Riverview Place, Metroplex on Gateway, Murarrie, QLD 4172, Australia; helenowenvet@hotmail.com; 6Biosecurity New Zealand, Ministry for Primary Industries, Wellington 6140, New Zealand; jalawneh@gmail.com

**Keywords:** predictive modeling, inter-pathologist agreement, glass slides, whole-slide images, bladder disease, concurrence, canine, feline, veterinary pathology

## Abstract

**Simple Summary:**

There is a common joke in pathology—put three pathologists in a room and you will obtain three different answers. This saying comes from the fact that pathology can be subjective; pathologists’ diagnoses can be influenced by many different biases, and pathologists are also influenced by the presence or absence of animal information and medical history. Compared to pathology, statistics is a much more objective field. This study aimed to develop a probability-based tool using statistics obtained by analyzing 338 histopathology slides of canine and feline urinary bladders, then see if the tool affected agreement between the test pathologists. Four pathologists diagnosed 25 canine and feline bladder slides and they conducted this three times: without animal and clinical information, then with this information, and finally using the probability tool. Results showed large differences in the pathologists’ interpretation of bladder slides, with kappa agreement values (low value for digital slide images, high value for glass slides) of 7–37% without any animal or clinical information, 23–37% with animal signalment and history, and 31–42% when our probability tool was used. This study provides a starting point for the use of probability-based tools in standardizing pathologist agreement in veterinary pathology.

**Abstract:**

Inter-pathologist variation is widely recognized across human and veterinary pathology and is often compounded by missing animal or clinical information on pathology submission forms. Variation in pathologist threshold levels of resident inflammatory cells in the tissue of interest can further decrease inter-pathologist agreement. This study applied a predictive modeling tool to bladder histology slides that were assessed by four pathologists: first without animal and clinical information, then with this information, and finally using the predictive tool. All three assessments were performed twice, using digital whole-slide images (WSI) and then glass slides. Results showed marked variation in pathologists’ interpretation of bladder slides, with kappa agreement values of 7–37% without any animal or clinical information, 23–37% with animal signalment and history, and 31–42% when our predictive tool was applied, for digital WSI and glass slides. The concurrence of test pathologists to the reference diagnosis was 60% overall. This study provides a starting point for the use of predictive modeling in standardizing pathologist agreement in veterinary pathology. It also highlights the importance of high-quality whole-slide imaging to limit the effect of digitization on inter-pathologist agreement and the benefit of continued standardization of tissue assessment in veterinary pathology.

## 1. Introduction

Inter-pathologist variation has been widely recognized across the human and veterinary medical fields [1,2,3,4]. A Fleiss’ kappa statistic of greater than 40% is generally deemed to be a fair to a good level of agreement, and greater than 75%, an excellent level of agreement [5]. There are many causes for inter-pathologist variation including the level of experience [6], sample quality and processing [7], and the organ system being examined. Lack of standardization in pathology report writing may be a further limitation to disease diagnosis in veterinary pathology [4].

Predictive models have been used extensively in human medicine, for example, to estimate disease probability in fields, such as cardiac exercise testing, and to estimate the risk of cardiac disease or lung cancer given multiple test results and patient factors [8,9,10,11]. In the veterinary field, predictive models have been used in genetics [12], ultrasonography [13,14], surgery and surgical prognosis development [15,16] as well as predicting disease outbreaks [17]; however, their use in veterinary histopathology has been limited thus far to occasional studies in wildlife [18]. Predictive logistic regression models have the potential to assist decision making in veterinary histopathological diagnosis and prognosis and improve pathologist and veterinarian awareness of disease risk in each individual patient.

Urinary bladder tissue was chosen as a pilot tissue for this study, as part of a doctoral research project. Broadly speaking, bladder diseases in dogs and cats fit one of three categories: neoplasia, urolithiasis (bladder stones, with or without a sterile or infectious cystitis) [19], cystitis (inflammation in the bladder wall without stones, which may be due to infectious or non-infectious causes), with a small number of sampled bladders having mild, non-specific other changes [20]. Cystitis and urolithiasis can show similar inflammatory changes and they require different treatments; however, the diagnosis of urolithiasis is typically based on the clinical presence of bladder stones. While standardization of pathology reporting for neoplasia in dogs and cats is a current area of research focus, some of which has recently been collated by the Veterinary Cancer Society’s Oncology-Pathology working group [21], there has been less focus on the inflammatory disease in these species. Clearly defined histological criteria improve pathologist agreement in the assessment and interpretation of histological changes [4,22]. There exists standardized terminology for use on urinary bladder tissues in rodent toxicologic pathology [23], however, to date, there are no standardized histological criteria for urinary bladder assessment in companion animals. This lack of standardization became apparent in a previous study by this research group, in which seven percent of retrospectively collected canine or feline bladder histology cases had their diagnosis changed, most commonly reclassifying a cystitis diagnosis as normal [20]. The discrepancy was commonly related to the number of leukocytes present, and the concurrent presence of hemorrhage and edema [20].

To compound inter-pathologist variability, incomplete relevant clinical information on pathology submission forms is a frequent problem in both human and veterinary pathology. In one veterinary study of 510 biopsy submissions, up to 88% of forms were deficient in at least one key area [24]. In human medicine, higher quality clinical information on pathology requisition forms is associated with decreased turnaround time (*p* < 0.001) and improved outcomes [25], while the absence of clinical history may result in lower diagnostic accuracy [26].

A rapidly growing facet of pathology is the use of digital pathology and the evaluation of digital whole-slide images (WSI) in lieu of glass slides. While conflicting views on the use of digital WSI remain, current literature suggests that WSI are comparable to glass slides in teaching, research and diagnostic settings [27,28,29,30,31], and the general consensus is leaning towards embracing the use of this tool in both teaching and diagnostic fields of veterinary pathology [29,32]. This study aimed to evaluate pathologist agreement and concurrence with the reference diagnosis when diagnosing bladder tissue samples without access to clinical history when provided with clinical history and with the use of a predictive tool. A secondary aim was to compare agreement between pathologists when evaluating glass slides versus digital WSI.

## 2. Materials and Methods

### 2.1. Sample Selection and Processing

The sample size for this project was calculated using the kappa sample size calculation. We assumed baseline agreement (κ_0_) = 0.5 [4], and assumed the lower limit of agreement (κ_L_) to be ≥0.2. The decision was made to involve boarded veterinary pathologists from four different countries (Canada, USA, Australia, and New Zealand) to provide a global perspective. Power calculations showed that to obtain statistical power with four pathologists evaluating the slides, at least 23 cases needed to be evaluated. The bladder tissue samples included in this study were selected from a bladder disease case material pool (EJ) [20] and processed routinely for H&E microscopy on glass slides. Slide selection criteria were that the sample was full thickness (contained outer muscularis with or without serosa), the tissue section on the microscope slide was at least 0.5 cm × 0.5 cm in size and oriented in a cross-section, and samples represented varying degrees of severity of each diagnosis group (cystitis, neoplasia, and urolithiasis) or represented normal bladder tissue for both dogs and cats. Twenty-five cases, consisting of seven cystitis (two canine and five feline), six neoplasia (three canine and three feline), six urolithiasis including one with concurrent urinary tract infection (four canine and two feline), and six normal bladders (four canine and two feline) met inclusion criteria. Slides were digitally scanned using a Leica Aperio CS2 slide scanner (serial number 50019) and ScanScope software, using a 40× objective and producing an image with a resolution of 0.25 μM/pixel. Examples of these images are represented in Appendix A. In the second component of this study, three glass slide recuts were made of each case (three neoplasia samples were unavailable so glass slide *n* = 22) and mailed to each study participant for viewing on their own microscope, 11 months after viewing the WSI.

### 2.2. Building the Predictive Tool

In a previous study, multinomial logistic regression modeling identified six significant variables that were associated with bladder disease diagnosis species, urothelial ulceration, urothelial inflammation, submucosal inflammation type, presence of lymphoid aggregates and amount of submucosal hemorrhage [20]. For the present study, a spreadsheet was built to record the findings, diagnoses, and comments from each pathologist evaluating the slide set under the three sequential conditions with no animal or clinical information, then with animal signalment and clinical history and finally with the adjunct use of the predictive tool (Figure 1). For the first (no signalment or history) and second (signalment and history available) worksheets, cells contained a drop-down list with yes or no options for the histological variables or a list of inflammatory cell types, except for the morphological diagnosis and comment columns which allowed free text responses (Table 1). For etiological diagnosis, pathologists could select one response from a drop-down list containing cystitis, neoplasia, urolithiasis, normal, or ‘other’. To reduce the complexity of the task and the subsequent statistical analysis, pathologists were asked to provide drop-down answers to selected histological features that were deemed to be representative of any disease process in the bladder–urothelial ulceration, submucosal edema, submucosal hemorrhage, presence of submucosal inflammation, the primary type of submucosal inflammation, presence of muscularis inflammation, the primary type of muscularis inflammation, and the presence of microorganisms.

The third worksheet, using the predictive tool with canine and feline samples on separate worksheets, was designed so that the probabilities for each disease for every possible combination of variables (derived from the logistic regression modeling) could be stored in a hidden worksheet. Pathologists recorded evaluations of the following criteria: urothelial ulceration, submucosal lymphoid aggregates, neutrophilic submucosal inflammation, urothelial inflammation, and amount of submucosal hemorrhage (Table 2). The worksheet table visible to the pathologists would populate in real time with the probability for each disease based on the combination of histological variables observed by the pathologist before the pathologist was prompted to make a diagnosis. In addition to displaying the probabilities and confidence intervals in a table, a chart was also set up to graphically represent the probabilities for each disease for every slide evaluated.

Notably, when cases were selected for this study by the reference pathologists, all cases were normal or diagnosed as urolithiasis, cystitis, or neoplasia by two readers (RA and EJ) with clinical information available, as described previously [20]. A category for ‘other’ diagnoses was included in the original logistic regression modeling (combined with normal to make the baseline category due to relatively low case numbers in both categories); therefore, the predictive tool provided probabilities for ‘normal/other’ combined. The ‘other’ diagnosis category was, therefore, included in this experiment, even though there were no cases belonging to this diagnosis category based on the reference diagnosis.

To summarize, our four study pathologists were asked to make a diagnosis based on their own individual criteria for the first two reads (without then with clinical history). Then, for the third read using the predictive tool, probabilities would be displayed based on their responses to certain histologic features (Table 1). Thus, the ultimate goal of the study was to evaluate the influence of patient signalment and history, glass slides vs. digital WSI, and the predictive tool on inter-pathologist variation and concurrence with the reference diagnosis.

### 2.3. Pathologists

Four veterinary pathologists were selected based on the following criteria: specialist certification with the American College of Veterinary Pathologists (ACVP), and a minimum of five years working in a diagnostic environment that involved the review of tissues from dogs and cats, and from a variety of geographical locations (United States of America, Canada, New Zealand, and Australia). Pathologists viewed the slides on the computer they use routinely for their work using Aperio’s eSlide Manager (version 12.4.0.5043, 2018). Pathologist results were deidentified and randomly assigned a number from one to four (P1–P4). Data were managed via Microsoft Office Excel [33].

### 2.4. Statistical Analysis

All analyses were conducted in R [34]. For each study component (glass slides and WSI), the final data from the four pathologists were combined to form a single file with a unique slide identifier (ID), species, conditions (no animal information, signalment and history, and predictive tool), and the diagnosis by each pathologist. The Fleiss kappa coefficient was used to compute the inter-rater reliability measures as an agreement measure for multiple categorical variables [5,35]. According to Fleiss et al. (2003), the kappa value can range from −1 (no agreement) to +1 (perfect agreement) with κ = 0 indicating that the agreement is no better than what would be obtained by chance [5]. Fleiss kappa values greater than 0.75 represent excellent agreement beyond chance, values between 0.40 and 0.75 represent fair to good agreement beyond chance and values below 0.40 or so may be taken to represent poor agreement beyond chance [5]. The accuracy (or concurrence) of a test refers to the ability of that test to give a true measure of the item being tested, with a highly accurate test having high sensitivity and specificity [36]. In general, accuracy is measured similarly to concordance or agreement, with >75% being an acceptable level of accuracy [5]. In order to validate the accuracy of each of the methods, sensitivity (the ability of a method to correctly classify an individual as having the “disease”) and specificity (the ability of a method to correctly classify an individual as “disease-free”) measures for each of the methods were computed against the reference diagnosis.

Pathologist number 4 (P4) had some technical issues that prevented their assessment of all the digital slides. The MICE R package was used for imputing the missing diagnoses from this pathologist. As the outcome here is a multinomial variable, the polytomous logistic regression imputation model for unordered categorical data was implemented by the MICE algorithm as described previously [37]. For the evaluation of pathologist concurrence with the reference diagnosis, cases for which any pathologist had made a diagnosis of ‘other’ could not be compared against the reference diagnosis and was removed.

## 3. Results

### 3.1. Data Overview

From the count data, it was evident that there was a high level of variation between the four pathologists throughout the intervention steps. The diagnosis for each case by each pathologist is shown in Table 3 and Table 4. For some cases there was good agreement between pathologists and each pathologist stayed consistent with their diagnosis, particularly for cases with neoplasia. However, there was wide variation in diagnoses both between pathologists and for the same pathologist throughout the sequential slide readings for many cases. This was particularly evident for slides with urolithiasis or cystitis, and for cases with normal bladder tissue.

### 3.2. Statistical Analysis of Inter-Pathologist Agreement

#### 3.2.1. Digital Whole-Slide Images

The Fleiss kappa (κ) overall measures of agreement were found to be 0.07 (*p* = 0.13), 0.237 (*p* < 0.01) and 0.31 (*p* < 0.001), respectively, for no animal information, signalment and history, and predictive tool conditions when evaluating digital whole-slide images (Table 5, Figure 2). According to Fleiss classification [5], the results show a poor level of inter-pathologist agreement across all slide reading conditions, and the 0.07 kappa for no animal information represents a statistically non-significant poor agreement between the four study pathologists.

For the first read with no animal information, there was good agreement for diagnosing neoplasia (κ = 0.56, *p* < 0.001), but poor agreement for all other diagnosis groups. For the second read where clinical history and signalment were made available, there was a fair to good agreement between the four pathologists when diagnosing cystitis (κ = 0.56, *p* < 0.001) and excellent agreement (κ = 0.77, *p* < 0.001) in rating patients as having neoplasia. There was poor agreement in rating patients as having normal bladder tissue, and non-significant poor agreement for urolithiasis.

Poor agreement (κ = 0.31) between the four pathologists was found when using the predictive probabilities diagnostic tool; however, the predictive tool had the highest overall agreement of the three slide reading conditions. When using the predictive tool there was good agreement (κ = 0.55, *p* < 0.001) in diagnosing cases as having neoplasia and a borderline fair to good agreement (κ = 0.39, *p* < 0.001) between the four pathologists rating cases as having normal bladder tissue; there was poor agreement in rating patients as having cystitis or urolithiasis.

#### 3.2.2. Glass Slides

For glass slides, agreement was found to be 0.37 (*p* < 0.001), 0.37 (*p* < 0.001) and 0.42 (*p* < 0.001), respectively, for no animal information, signalment and history, and predictive tool conditions when evaluating the glass slides (Table 6, Figure 3). These results show higher agreement in all slide reading conditions when compared to the digital whole-slide images; however, there is still a poor level of agreement for the no information and signalment and history conditions. The use of the predictive tool resulted in a fair to good level of agreement.

For the first read with no animal information, there was fair to good agreement for diagnosing neoplasia (κ = 0.69, *p* < 0.001) and normal bladder tissue (κ = 0.60, *p* < 0.001) but poor agreement for the other diagnosis groups. For the second read where clinical history and signalment was made available, there was fair to good agreement between the four pathologists when diagnosing cystitis (κ = 0.69, *p* < 0.001), normal bladder tissue (κ = 0.55, *p* < 0.001), and neoplasia (κ = 0.69, *p* < 0.001). There was poor agreement in rating patients as having urolithiasis.

Borderline fair to good overall agreement (κ = 0.42) between the four pathologists was found when using the predictive probabilities diagnostic tool, meaning the predictive tool had the highest overall agreement of the three slide reading conditions. When using the predictive tool there was fair to good agreement in diagnosing cases as having neoplasia or cystitis or being normal bladder tissue, while there was a poor agreement for urolithiasis diagnosis.

### 3.3. Evaluation of Concurrence of Pathologist Diagnosis with the Reference Diagnosis

The overall data showed a concurrence of 0.60 by the four pathologists when compared to the reference diagnosis. When no animal information was available, the concurrence was 0.55, while the addition of animal signalment and history increased the concurrence to 0.61. The predictive probability tool showed an improved concurrence of 0.65 compared to the other slide reading conditions. Diagnosis using digital whole-slide images had a concurrence of 0.58 (*p* < 0.001), while the evaluation of glass slides in this study displayed an overall concurrence of 0.63.

The proportion of agreement with the reference diagnosis using the “no animal information” approach was 0.38, therefore, 62% of the agreement was due to chance. The overall data and the other approaches indicated >40% agreement that is not due to chance. Greater than 50% agreement was observed using the predictive probability tool that is not attributable to chance alone (Table 7 and Figure 4).

The sensitivity and specificity of the diagnostic approaches used by the four test pathologists are presented in Appendix A. Overall, a diagnosis of cystitis had the highest sensitivity (70%) but lowest specificity (70%) of all the diagnoses. The sensitivity of diagnosing urolithiasis was highly influenced by the additional clinical history. When comparing glass slides to digital slide images, cystitis diagnosis had higher sensitivity and lower specificity with glass slides; neoplasia had a higher sensitivity and comparable specificity; normal had higher sensitivity and specificity with glass slides and the urolithiasis diagnosis had lower sensitivity but higher specificity when using glass slides compared to digital slide images.

## 4. Discussion

Histopathology forms the majority of a veterinary pathologist’s workload, in fact, expertise in histopathological interpretation is one of the unique skill sets of the specialty. However, it has been widely demonstrated that pathologists fall victim to various processing factors and individual cognitive biases that can impact their diagnoses [6,7]. The goals of this study were to investigate inter-pathologist agreement when evaluating canine and feline bladder histology, to evaluate the effect of a predictive tool on inter-pathologist agreement, and to evaluate pathologist concurrence with the author-generated reference diagnosis. A secondary goal of this study was to compare the inter-pathologist agreement and concurrence with the reference diagnosis when diagnosing the same cases on digital whole-slide images compared to glass slides. Bladder tissue was chosen as the test tissue as it has simple anatomy and limited ways in which it can respond to injury, thus it was deemed an appropriate tissue to trial a novel approach, such as this.

Pathologist agreement improved markedly throughout the slide reads for both digital whole-slide images and glass slides particularly without and with animal information and clinical history, highlighting the difficulty of interpreting histology without this information, and unfortunately, a common problem encountered by veterinary pathologists working in commercial diagnostic laboratories. Despite sometimes being viewed as biasing the pathologist, the norm in toxicologic pathology studies is to provide this information; Crissman and colleagues state that “if this information is not available initially, selected tissues may need to be re-evaluated to ensure accurate diagnoses and interpretations” [38]. The predictive tool did have a small effect on inter-pathologist agreement, increasing pathologist agreement compared to signalment and history from 23% to 31% (digital WSI) and from 37% to 42% (glass slides). Interestingly, there was a negative effect of the predictive tool on the diagnosis of cystitis for both digital WSI and glass slides, with lower agreement for cystitis diagnosis with the predictive tool compared to the animal signalment and clinical history. This could be attributed to the subjective and variable interpretation of subtle histological features, such as leukocytic infiltrates by individual pathologists, causing the predictive tool to display higher or lower probabilities for cystitis for some pathologists compared to others. It is important to note, that for the glass slides, each pathologist received a recut of the block and thus may have received a slightly different region of tissue to evaluate (the four sections were cut at 4 microns width and sequentially; therefore, the maximum difference between the four slides is the width of two lymphocytes at 16 um), whilst the digital images were identical for all pathologists. In addition, there is frequently overlap in histological features in urinary bladder diseases, particularly between cases of cystitis and urolithiasis, which can lead to variations in diagnoses between pathologists. It is possible that subtle differences in the slides resulted in variable diagnoses of cystitis versus urolithiasis for each pathologist; for example, when the glass slide results for urolithiasis and cystitis diagnoses are combined, there are 13 in total (reference diagnosis), with test pathologists all diagnosing 14 or 15 in total, thus the agreement may be higher if these two categories were combined. On a related note, the limited number of categories meant that test pathologists were not afforded the flexibility to select a diagnosis outside of those provided which may have impacted our results, although some recent work in human pathology suggests that inter-pathologist agreement decreases with an increasing number of potential categories [39].

There were several slides where the pathologist altered their diagnosis once or twice throughout the experiment. This occurred most between diagnosis of cystitis or normal, with borderline poor to fair agreement for the diagnosis of normal bladder tissue. Some pathologists commented that they called a case ‘cystitis’ even though the inflammation was minimal and/or likely clinically insignificant. This could be explained by the concept of context bias, the tendency to call a sample abnormal when viewed alongside other samples with a high disease prevalence [40,41]. A comment from one of the test pathologists also raised the fact that some pathologists may inadvertently diagnose a false-positive result due to a feeling of wanting to provide some kind of answer to the clinician. This finding also raises the question of what encompasses normal leukocyte numbers for urinary bladder in companion animals, as there is no clear current consensus. While there are few publications on normal urothelial lymphocyte numbers, with normal human urothelium and submucosa containing up to 42 lymphocytes per field (size unspecified but interpreted by the author as a maximum of 150× magnification) [42], the authors have been unable to find published information on the normal numbers of submucosal resident lymphocytes in the bladder wall of dogs and cats. In a previous study on canine and feline bladder histology, a tissue was assigned a diagnosis of ‘normal’ if the submucosa contained up to 20 lymphocytes per low power (100×) field, without hemorrhage and edema. These leukocytes were deemed to be normal resident lymphocytes. If a section contained greater than 20 lymphocytes per low power field or had up to 20, but had a concurrent vascular reaction (hemorrhage and/or edema), or had any neutrophil infiltration, then the tissue was classified as inflamed and assigned to the cystitis group [20]. One pathologist commented in the free text that they classified suspected cases of feline lower urinary tract obstruction (based on the clinical history) as urolithiasis, as they deemed the mucous plugs to have a similar physiological effect, whereas other pathologists classified this type of clinical history and histological change as cystitis. Future work using this data will include an in-depth analysis of the free text morphological descriptions and comments. This work also highlights that standardized case criteria from consensus groups of pathologists would be beneficial for domestic animals, addressing various pathologies in common organs.

The second component of the statistical analysis evaluated the concurrence of each pathologist with the reference diagnosis. Our finding of increasing agreement with the reference diagnosis when clinical history was made available, then further increasing agreement when the predictive tool was applied suggests that there could be some value in the use of such a tool, particularly once confounding variables, as discussed below, have been resolved. It was interesting to note that the predictive tool did improve inter-pathologist agreement but had only a minimal effect on concurrence with the reference diagnosis. This could be explained by the fact that the pathologists changed their diagnosis fairly consistently throughout the slide reading conditions; for example, for a cystitis case, P1 diagnosed cystitis and P2 diagnosed normal then with the next slide-reading condition P1 diagnosed normal and P2 diagnosed cystitis—these have a poor inter-rater agreement but 50% concurrence with the reference diagnosis. One limitation of this study is the formulation of the reference diagnosis by one board certified pathologist and one pathology trainee, albeit in isolation, then a consensus was reached. For cases that had marked variation in diagnosis, it is possible that the reference diagnosis was incorrect and one or more of the test pathologists were correct. For future studies of this nature, a working group of multiple specialist pathologists would be of value in determining the reference diagnoses, or, at minimum, two specialist pathologists to decide on the reference diagnoses. Working groups are the current gold standard used in fields, such as toxicologic pathology, and oncology harmonization [43,44,45].

One major limitation of this study is that in attempting to encompass all possible bladder wall histological lesion patterns in the initial regression model, cases with a diagnosis of urolithiasis were included. The traumatic injury and inflammation associated with the stone can be similar to that encountered in sterile or infectious cystitis making urolithiasis a clinical diagnosis, not a histological one. Urolithiasis cases were sometimes found to have different changes to that of cystitis (more submucosal hemorrhage and edema, and less inflammation) in our initial modeling [20], however, more work is required in this area. The fact that the study pathologists were not able to differentiate between cystitis and urolithiasis based on histology alone was an expected finding and highlights the importance of accurate clinical information on pathology submission forms [24].

The diagnostic variation between some of the test pathologists could be explained by several factors. Firstly, free text comments from all pathologists described difficulty in differentiating inflammatory cell types on the digital images. This is known to be an issue with digitized slides and can be improved by a number of factors including thinner sections (three micron sections were ideal in one study) [46] and further optimization of scanner settings for the tissue and staining characteristics of each individual laboratory [46]. Digitally scanned microscope slides are becoming more commonplace in veterinary pathology [29,47,48]; however, it is vital that the imaging is validated to ensure the same, if not better, diagnostic performance as glass slides [49]. The College of American Pathologists Pathology and Laboratory Quality Center recommends validation on whole-slide images using at least 60 routine cases and comparing intra-observer agreement between the digital and glass slides viewed at least two weeks apart [49]. Inter-pathologist agreement increased from fair to moderate and concurrence increased, but still remained low (<75%) in this study when analyzing the digital and glass slide data, respectively; however, this is difficult to interpret without the recommended validation phase. In future studies it would be useful to validate the scanned images and include a familiarization period for pathologists using the digital software. Secondly, there was some variation in the individual interpretation of the instructions by some pathologists; this could have been mitigated by a shared training session.

No cases with ‘other’ diagnoses were included in this study, so it was interesting to observe the frequency with which our study pathologists selected this option. It is known that when provided with an ‘intermediate’ option for tumor grading, pathologists tend to most frequently diagnose the intermediate category and avoid the extreme ranges [50], so what we see here may be a similar phenomenon. It is also possible that the diagnosis of ‘other’ by our study pathologists was made as a last resort when they did not think their diagnosis fitted exactly into one of the other categories, as was found in a consensus review of human breast pathology cases [51]. Most of the ‘other’ diagnoses in our dataset were cases of cystitis or normal bladder and could be attributed to the reasons described above; an unclear definition of normal bladder leukocyte numbers, an unclear definition of histological features of some bladder diseases, such as feline idiopathic cystitis, and avoidance of extreme scoring ranges, pathologists tend to avoid extreme scoring ranges including the lowest severity grading ranged for lesions, i.e., normal tissue [40,52].

The literature suggests that clearly defined histological criteria improve pathologist agreement regarding the assessment and interpretation of histological changes [4,22]. Our application of a predictive probability tool is the first step toward providing a standardized criterion for the evaluation of bladder disease in dogs and cats, and the findings of this study can be extrapolated to human histopathology studies. Our preliminary analysis suggests that the tool had no significant effect on pathologist agreement; however, it did improve concurrence in comparison to the reference diagnosis. These findings indicate that further research is required in this area, particularly to explore other histological contexts and diagnoses, as well as to explore methodological confounders in this study including the impact of reading digital scans, and potential variation in interpretation of instructions. Our findings also imply that as predictive tools are developed to standardize diagnoses and predict prognoses in veterinary diagnostic contexts, that training on the application may be needed to maximize the accurate use of the tool.

## 5. Conclusions

In conclusion, we found good levels of agreement between four veterinary pathologists evaluating canine and feline bladder sections to diagnose bladder neoplasia; however, there was poor to fair agreement for the diagnosis of cystitis, urolithiasis, and normal bladder tissue. Agreement between pathologists improved when signalment and clinical history were provided, with mixed results when a predictive probability tool was added. The predictive tool did prove valuable in increasing the concurrence of the pathologists’ diagnosis to the reference diagnosis; however, further exploration of this area is warranted using a dataset more representative of the prevalence of these diseases in the general cat and dog population. Further research is needed to determine appropriate parameters for inflammatory cells in normal canine and feline bladder tissue.

## Figures and Tables

**Figure 1 vetsci-09-00367-f001:**

Sequence of assessment of the slide set by each pathologist.

**Figure 2 vetsci-09-00367-f002:**
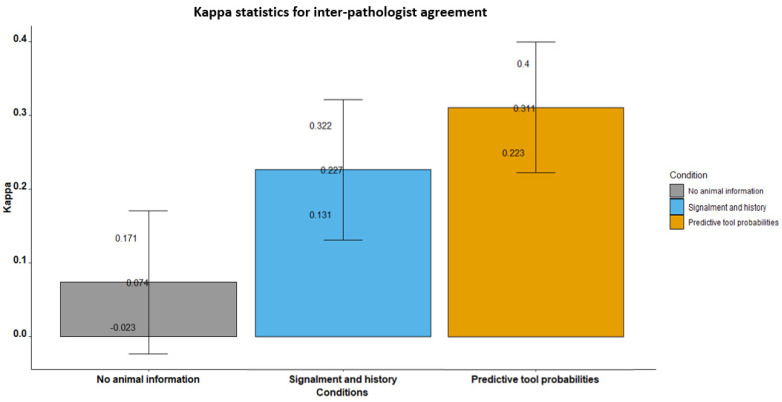
Bar plot showing the inter-pathologist agreement kappa statistics with 95% CI for the three slide-reading conditions, diagnosing digital slides of canine and feline bladder tissue.

**Figure 3 vetsci-09-00367-f003:**
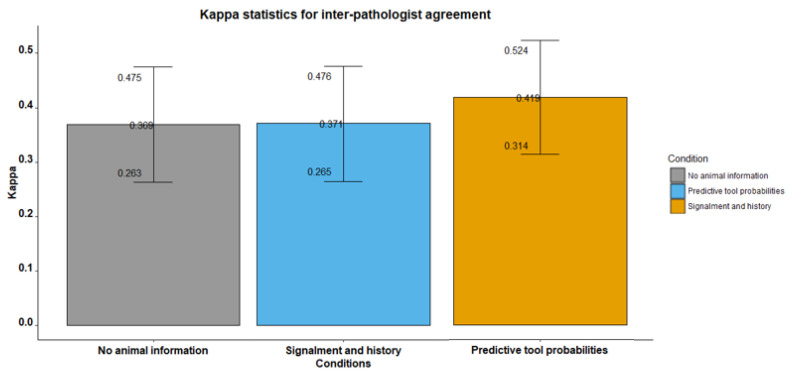
Bar plot showing the inter-pathologist agreement kappa statistics with 95% CI for the three slide-reading conditions, diagnosing glass slides of canine and feline bladder tissue.

**Figure 4 vetsci-09-00367-f004:**
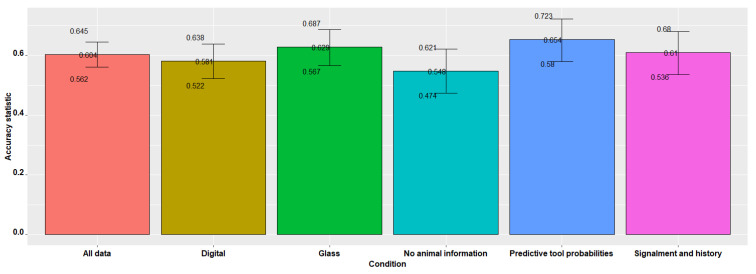
Concurrence of the four pathologists’ diagnoses of canine and feline bladder tissues compared with the reference diagnosis.

**Table 1 vetsci-09-00367-t001:** Histological criteria to be assessed by the pathologists in worksheets one and two (without and with signalment and clinical history).

Column Heading	Potential Answers *
Slide code	Provided
Ulceration	Yes, No
SM_oedema	Yes, No
SM_haem	Yes, No
SM_inflamm	Yes, No
SM_inflamm_type	Lymphocytic Lymphoplasmacytic Neutrophilic Granulomatous No inflammation
Det_inflamm	Yes, No
Det_inflamm_type	Lymphocytic Lymphoplasmacytic Neutrophilic Granulomatous No inflammation
Organisms	Yes, No
Morphological diagnosis	Free form box
Etiological diagnosis	Normal Other Cystitis Neoplasia Urolithiasis
Comments	Free form box

Det detrusor muscle/muscularis; haem hemorrhage; inflamm inflammation; SM submucosal. * Potential answers provided from a drop-down box; no free text allowed unless otherwise stated.

**Table 2 vetsci-09-00367-t002:** Histological criteria to be assessed by the pathologists in worksheets three and four (canine and feline), using the predictive tool.

Column Heading	Potential Answers *
Slide code	Provided
Urothelial ulceration	Yes, No
Submucosal lymphoid aggregates	Yes, No
Neutrophilic submucosal inflammation	Yes, No
Urothelial inflammation	Yes, No
Amount of submucosal hemorrhage	Mild Moderate Severe
Your diagnosis	Normal Other Cystitis Neoplasia Urolithiasis
Comments	Free form box

* Potential answers provided from a drop-down box; no free text allowed unless otherwise stated.

**Table 3 vetsci-09-00367-t003:** Digital whole-slide image count data from all study pathologists, P1–P4.

		No Animal Information				Signalment and History				With Predictive Tool			
Diagnosis	Reference	P1	P2	P3	P4	P1	P2	P3	P4	P1	P2	P3	P4
Cystitis	7	7	14	17	6	6	14	11	5	7	14	11	4
Neoplasia	6	4	4	3	3	4	4	3	3	4	4	3	3
Urolithiasis	6	9	0	1	6	9	0	8	5	7	0	8	3
Normal	6	5	2	2	1	6	2	2	5	5	2	2	5
Other	0	0	5	2	3	0	5	1	3	2	5	1	4
Total	25	25	25	25	19 *	25	25	25	21 *	25	25	25	17 *

* Technical issues prevented P4 from viewing some slides.

**Table 4 vetsci-09-00367-t004:** Glass slide count data from all study pathologists, P1–P4.

		No Animal Information				Signalment and History				With Predictive Tool			
Diagnosis	Reference	P1	P2	P3	P4	P1	P2	P3	P4	P1	P2	P3	P4
Cystitis	7	6	15	15	10	6	15	9	10	6	14	11	11
Neoplasia	6	3	3	2	3	2	3	3	3	3	3	3	3
Urolithiasis	6	8	0	1	3	9	0	4	3	8	0	4	1
Normal	6	5	2	4	4	4	0	4	4	5	2	4	4
Other	0	0	2	0	2	0	0	2	2	0	3	0	3
Total	25 *	22	22	22	22	21 **	22	22	22	22	22	22	22

* Three blocks from the WSI part of the study were unavailable for sectioning glass slides. ** No data recorded for one case in the first spreadsheet.

**Table 5 vetsci-09-00367-t005:** Inter-pathologist agreement for the three slide-reading conditions, diagnosing digital whole-slide images of canine and feline bladder tissue.

	Inter-Pathologist Agreement: Fleiss Kappa Statistics					
	Overall Kappa			Detailed Kappa for Each Diagnosis		
	Kappa	Z-Value	*p*-Value	Kappa	Z-Value	*p*-Value
No animal information						
overall	0.074	1.5	0.134			
cystitis				0.01	0.118	0.906
neoplasia				0.558	6.833	<0.001
normal				0.204	2.501	0.012
other				−0.02	−0.25	0.803
urolithiasis				−0.159	−1.943	0.052
Signalment and history						
overall	0.227	4.668	<0.001			
cystitis				0.558	6.833	<0.001
neoplasia				0.765	9.366	<0.001
normal				0.268	3.278	0.001
other				−0.01	−0.124	0.902
urolithiasis				0.049	0.604	0.546
Predictive tool probabilities						
overall	0.311	6.873	<0.001			
cystitis				0.204	2.501	0.012
neoplasia				0.551	6.75	<0.001
normal				0.391	4.788	<0.001
other				0.054	0.666	0.505
urolithiasis				0.307	3.755	<0.001

Z = standard normal score.

**Table 6 vetsci-09-00367-t006:** Inter-pathologist agreement for the three slide-reading conditions, diagnosing glass slides of canine and feline bladder tissue.

	Inter-Pathologist Agreement: Fleiss Kappa Statistics					
	Overall Kappa			Detailed Kappa for Each Diagnosis		
	Kappa	Z-Value	*p*-Value	Kappa	Z-Value	*p*-Value
No animal information						
overall	0.369	6.813	<0.001			
cystitis				0.362	4.163	<0.001
neoplasia				0.688	7.908	<0.001
normal				0.604	6.935	<0.001
urolithiasis				−0.045	−0.517	0.605
Signalment and history						
overall	0.371	6.901	<0.001			
cystitis				0.688	7.908	<0.001
neoplasia				0.688	7.908	<0.001
normal				0.545	6.257	<0.001
urolithiasis				0.152	1.741	0.082
Predictive tool probabilities						
overall	0.419	7.84	<0.001			
cystitis				0.604	6.935	<0.001
neoplasia				0.678	7.794	<0.001
normal				0.652	7.488	<0.001
urolithiasis				0.127	1.464	0.143

Z = standard normal score.

**Table 7 vetsci-09-00367-t007:** Concurrence of the four pathologists’ diagnoses of canine and feline bladder tissues compared with the reference diagnosis.

Concurrence and Kappa Statistics					
Concurrence				Agreement	
Concurrence	LCL	UCL	*p*-Value	Kappa	*p*-Value
All data					
0.604	0.562	0.645	<0.001	0.460	<0.001
No animal information					
0.548	0.474	0.621	<0.001	0.384	<0.001
Signalment and history					
0.610	0.536	0.680	<0.001	0.470	0.002
Predictive tool probabilities					
0.654	0.580	0.723	<0.001	0.528	0.001
Glass					
0.629	0.567	0.687	<0.001	0.486	<0.001
Digital					
0.581	0.522	0.638	<0.001	0.436	<0.001

LCL lower confidence limit; UCL upper confidence limit.

## Data Availability

Not applicable.

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
