# Peer review of "Evaluation of a Probability-Based Predictive Tool on Pathologist Agreement Using Urinary Bladder as a Pilot Tissue"

_vetsci, 2022, doi:10.3390/vetsci9070367_

Round 1
Reviewer 1 Report
This study evaluated the diagnostic concordance of veterinary pathologists using conventional cyst specimens, whole slide imaging, and a in-house diagnostic prediction tool. The conclusions drawn in this study are logically valid, such as tumors with presumably clear diagnostic criteria have a high rate of diagnostic concordance, whereas nonneoplastic lesions have a low rate. However, this paper contains the following issues
A brief summary of the significance and historical aspects of the tool would be desirable.
While it would be meaningful to diagnose tumors in pathology, an explanation is needed as to why it is necessary to determine the difference between cystitis and urolithiasis, which are non-neoplastic conditions. The impact of these diagnostic differences on treatment. should be mentioned.
Lines 243 and 298. Please confirm that “Error! Not a valid bookmark self-reference” is correct.
Tables 5 and 6. It is confusing that the Overall Kappa value is listed in the cystitis row; is it necessary to separate Overall and Detailed into separate columns?
All abbreviations need to be spelled out the first time they are used, as there is no sufficient explanation for some abbreviations, i. e., LCL, etc.
This paper should have examined the impact of predictive tools and WSI on inter-diagnostic variability, but there is no mention of WSI in the concluding chapter.
Author Response
A brief summary of the significance and historical aspects of the tool would be desirable. This is good advice, I have included this in the introduction lines 54-65.
While it would be meaningful to diagnose tumors in pathology, an explanation is needed as to why it is necessary to determine the difference between cystitis and urolithiasis, which are non-neoplastic conditions. The impact of these diagnostic differences on treatment. should be mentioned. Thank you for this good point, I have added some information on this this on lines 66-75.
Lines 243 and 298. Please confirm that “Error! Not a valid bookmark self-reference” is correct.
Thank you, this has been corrected.
Tables 5 and 6. It is confusing that the Overall Kappa value is listed in the cystitis row; is it necessary to separate Overall and Detailed into separate columns? I understand this concern as I initially agreed with you, however over time I became convinced that it is much easier to read the overall kappa, but having the detailed ones are important to show the variation between the different diagnoses. Thank you for this feedback, I have added an additional row in each section for the overall statistics which I feel has helped improve the clarity.
All abbreviations need to be spelled out the first time they are used, as there is no sufficient explanation for some abbreviations, i. e., LCL, etc. This has been corrected.
This paper should have examined the impact of predictive tools and WSI on inter-diagnostic variability, but there is no mention of WSI in the concluding chapter. Thank you for this feedback. I have discussed the limitations associated with WSI in this study in the paragraph starting on line 515, with discussion of agreement between WSI and glass slides from line 530, are you looking for more discussion than this?
Reviewer 2 Report
In this interesting manuscript, the authors propose an innovative predictive tool that could be used to distinguish between bladder neoplasia, cystitis, urolithiasis, and normal bladder tissue. This predictive tool could help to better diagnose bladder pathologies using histology slides, with or without clinical information. The authors found good levels of agreement between the four veterinary pathologists evaluating bladder sections to diagnose neoplasia, but not cystitis, urolithiasis, and normal bladder tissue. Agreement between pathologists improved when clinical history was provided. This predictive tool proved valuable in increasing the concurrence of the pathologists’ diagnosis to the reference diagnosis.
The predictive tool proposed in this manuscript is useful for the diagnosis of the canine and feline bladder pathologies, but its significance would highly increase if the results obtained using this innovative tool will be extrapolated to human bladder pathologies.
I consider that this manuscript is good, but it can be further improved. For example, in the Introduction Section, the authors could briefly discuss the main characteristics of bladder pathologies (neoplasia, cystitis, and urolithiasis). In the supplemental material of the manuscript, the authors could add at least a Figure with several representative digital whole-slide images as well as representative glass slides that were evaluated by the four pathologists.
The authors should explain all the abbreviations used in this manuscript, for example, LCL, and UCL.
Nevertheless, some minor revisions are required before publication, for example in Row 63-64, Row 154, Row 243, Row 298, Row 343-345, Row 440, etc.
Author Response
The predictive tool proposed in this manuscript is useful for the diagnosis of the canine and feline bladder pathologies, but its significance would highly increase if the results obtained using this innovative tool will be extrapolated to human bladder pathologies. Thank you for this feedback, I agree that the use of a similar tool could be of value to human bladder pathology. For now, I will have to consider this for future studies due to time constraints. I have mentioned this potential in the discussion, on line 557-558.
I consider that this manuscript is good, but it can be further improved. For example, in the Introduction Section, the authors could briefly discuss the main characteristics of bladder pathologies (neoplasia, cystitis, and urolithiasis). In the supplemental material of the manuscript, the authors could add at least a Figure with several representative digital whole-slide images as well as representative glass slides that were evaluated by the four pathologists. I like the idea of including some more pictures, thank you. I have included some representative whole slide images in the supplementary material and referred to this on line 113-114. I like the idea of introducing the bladder pathologies, I have added this to the introduction (lines 66-75).
The authors should explain all the abbreviations used in this manuscript, for example, LCL, and UCL. This has been corrected.
Nevertheless, some minor revisions are required before publication, for example in Row 63-64, Row 154, Row 243, Row 298, Row 343-345, Row 440, etc. Thank you very much for this helpful feedback. I was unable to find the concern in row 440.
Reviewer 3 Report
This work aims to reduce inter-pathologist variation in recognizing non normal tissues given the lack of standardization in pathology report writing. Indeed it can be identified another important aim that is the agreement between the reference and the pathologist diagnosis when evaluating glass slides versus digital whole slide images. The method that authors used is to take a pilot tissue, normal and pathological canine and feline bladder tissue in the case, and make it analyze by veterinary pathologists from different countries in three different ways: without access to clinical history, with clinical history and with a use of the predictive tool. The article finds out that there is a more important agreement between pathologists in the diagnosis of bladder neoplasia using the predictive tool, however there is a poor agreement in diagnosis concerning the inflammatory disease and in identifying normal bladder tissue. Further research is needed.
My suggestions:
Keywords should contain “canine” and “feline”
The cases evaluated by each pathologist are a number of 25 that is not a large amount of cases, in my opinion the study should be enlarged to a bigger number of samples to increase the variability and have a truer reflection of the agreement between pathologists
There are a lot of diagnostic tecniques in order to discriminate between oncologic and inflammatory bladder diseases and obtain a more certain reference diagnosis to compare with; for this reason I can suggest the analysis of this work https://pubmed.ncbi.nlm.nih.gov/34999017/
Clinical informations referring to the samples may confuse the pathology, in some cases they can help for the diagnosis but in some cases they can influence the valutation instead. In my opinion a pathology report writing is a trascription of an objective observation and additional informations can steer to a influenced conclusion.
Referring to lines 395 – 398 the fact that the glass slides sent to the various pathologists are not the same but different slides from a single sample, can increase the variability between diagnosis, distorting the whole work
Referring to lines 432 – 439 there are too little evidence about the normal number of submucosal limphocytes to make a diagnosis of cystitis based only on this evidence
There is a big overlap in the diagnosis of cystitis and urolithiasis due to the inflammatory nature of the two diseases. Authors should find more precise criteria and parameters to differentiate the two pathologic conditions
Author Response
My suggestions:
Keywords should contain “canine” and “feline” Thank you, these have been added.
The cases evaluated by each pathologist are a number of 25 that is not a large amount of cases, in my opinion the study should be enlarged to a bigger number of samples to increase the variability and have a truer reflection of the agreement between pathologists. I appreciate this comment as the number does seem small, however the kappa sample size calculation did suggest 23 slides minimum for 4 pathologists. I am happy to include the sample size calculation if you think it would benefit the manuscript.
There are a lot of diagnostic tecniques in order to discriminate between oncologic and inflammatory bladder diseases and obtain a more certain reference diagnosis to compare with; for this reason I can suggest the analysis of this work https://pubmed.ncbi.nlm.nih.gov/34999017/ This is an interesting paper, thank you for the recommendation. I agree that this kind of diagnosis would be very helpful, and this kind of technology is slowly becoming available for veterinary species, but not to the level that it would replace histological diagnosis.
Clinical informations referring to the samples may confuse the pathology, in some cases they can help for the diagnosis but in some cases they can influence the valutation instead. In my opinion a pathology report writing is a trascription of an objective observation and additional informations can steer to a influenced conclusion. This is common feedback and something I endeavoured to discuss in the manuscript, I have added more on this lines 383-387.
Referring to lines 395 – 398 the fact that the glass slides sent to the various pathologists are not the same but different slides from a single sample, can increase the variability between diagnosis, distorting the whole work I discussed this in the discussion, although the variability should be minimal as the 4 slides were all made at the same time therefore there is only 16um difference between the first and last slide (maximum width of two lymphocytes) therefore this was deemed to have had a negligible effect on the results.
Referring to lines 432 – 439 there are too little evidence about the normal number of submucosal limphocytes to make a diagnosis of cystitis based only on this evidence This is one of the main points of this manuscript – that in veterinary species we do not have clearly defined histological criteria. I am open to suggestions - do you have any suggestions on how else to do this histologically? The decisions made in this study are explained further in our previous publication which is cited, I can explain that more in this manuscript if necessary.
There is a big overlap in the diagnosis of cystitis and urolithiasis due to the inflammatory nature of the two diseases. Authors should find more precise criteria and parameters to differentiate the two pathologic conditions Thank you for giving me the opportunity to explain this further. I made sure to include in the discussion both the reason why urolithiasis was included as a separate diagnosis, and the acknowledgement that this is not typically a histological diagnosis. For similar research in future, I would plan to include only diseases for which there is a histological diagnosis, however unfortunately it was unavoidable in this preliminary study due to decisions made early on. I have added some disease definitions to the introduction (lines 54-60).
Reviewer 4 Report
I have read your manuscript“Evaluation of a probability-based predictive tool on pathologist agreement using urinary bladder as a pilot tissue” with great pleasure. This is an interesting article.
I have several comments:
1. This study points out that the use of predictive models in veterinary pathology improves the consistency of judgments, but does not seem to improve the accuracy of judgments, please discuss.
2. In Tables 5 and 6, the presentation of the overall kappa statistic results is easily misunderstood (it appears in the cystitis item), please revise the table.
3. In line 242-243, what is “Error! Not a valid bookmark self-reference “?
4. In line 298, what is “Error! Not a valid bookmark self-reference“?
5. Please note the formatting of lines 343-345 of the article.
Author Response
I have several comments:
- This study points out that the use of predictive models in veterinary pathology improves the consistency of judgments, but does not seem to improve the accuracy of judgments, please discuss. Thank you for giving me the opportunity to explain this more, I have elaborated on this in lines 484-492.
- In Tables 5 and 6, the presentation of the overall kappa statistic results is easily misunderstood (it appears in the cystitis item), please revise the table. Thank you for this feedback, I have added an additional row in each section for the overall statistics.
- In line 242-243, what is “Error! Not a valid bookmark self-reference“? Thank you, this has been corrected.
- In line 298, what is “Error! Not a valid bookmark self-reference“? Thank you, this has been corrected.
- Please note the formatting of lines 343-345 of the article. Thank you, this has been corrected.
Round 2
Reviewer 1 Report
The authors properly responded the comments of the reviewer and the revised manuscript has been improved for publication.
Reviewer 3 Report
Authors answered all comments and suggestions.
Reviewer 4 Report
The current form of the manuscript is acceptable.